# Synoptic and Seasonal Variability of Small River Plumes in the Northeastern Part of the Black Sea

Evgeniya Korshenko [1,2,*], Irina Panasenkova [1,2], Alexander Osadchiev [2,3], Pelagiya Belyakova [4] and Vladimir Fomin [1,5]

1   N.N. Zubov State Oceanographic Institute, Roshydromet, Moscow 119034, Russia
2   Shirshov Institute of Oceanology, Russian Academy of Sciences, Moscow 117997, Russia
3   Moscow Institute of Physics and Technology, Dolgoprudny 141701, Russia
4   Water Problems Institute, Russian Academy of Sciences, Moscow 119333, Russia
5   Marchuk Institute of Numerical Mathematics, Russian Academy of Sciences, Moscow 119333, Russia
*   Correspondence: zhenyakorshenko@gmail.com

**Abstract:** Small river plumes are typical features at many coastal regions in the World Ocean. These water masses have relatively small areas and volumes; however, due to their energetic dynamics localized in a thin surface layer, they strongly affect coastal circulation, water quality, and ocean-atmosphere interaction. In this study, we investigate external factors, which govern synoptic and seasonal variability of small river plumes, and, therefore, affect land-ocean fluxes of fluvial water and biogeochemically important material. We use numerical modeling to simulate small river plumes at the northeastern part of the Black Sea. We describe the response time of small river plumes to changes in river discharge and wind forcing conditions, which determines variability of river plumes at different time scales. We reveal that the influence of river plumes on coastal processes depends not only on total annual river discharge volume, but also on temporal distribution of high-discharge and low-discharge periods. Seasonal and synoptic features of local atmospheric circulation could strongly modify the relation between river plume characteristics and river discharge rate. The results obtained in this study are important for better assessment of delivery and fate of river-borne suspended and dissolved matter, as well as floating litter in coastal areas.

**Keywords:** river plume; spring-summer freshet; rain-induced flood; wind forcing; coastal circulation; water quality; Black Sea

## 1. Introduction

Most scientific works devoted to river plumes consider plumes formed only by large rivers, while river plumes formed by small rivers receive much less attention. This can be explained by the relatively small effect of an individual small river plume on the surrounding sea compared to a large river plume. In addition, small river plumes are characterized by a fast response (of an order of hours and days) to the variability of external influence [1–3] due to their small horizontal and vertical scales, which complicates in-situ measurements in them.

The relevance of studying small river plumes is determined both by their influence on regional processes in the coastal zone and by their role in global water transport from land to the World Ocean. Rivers, whose drainage basins do not exceed 10,000 km$^2$, account for 25% of water runoff and 40% of sediment runoff into the World Ocean [4,5]. In many coastal areas of the World with certain climatic conditions and coastline shapes, discharges of small rivers increase sharply during heavy rainfall periods. Therefore the total flow of small rivers becomes comparable to the flow of large rivers on a regional scale [6–9].

One such region is situated in the northeastern part of the Black Sea. Numerous gorges with steep slopes, located between the spurs of the Greater Caucasus, form drainage basins of the rivers flowing in the northeastern part of the Black Sea. Due to the heavy

dissection of the mountainous relief, the areas of these watersheds are relatively small (50–1500 km$^2$) [10]. These rivers are predominantly fed by seasonal snowmelt and rains. The steep slopes of drainage basins (up to 40–60°) of these rivers, their small size, and the high density of the river network lead to a rapid rainwater inflow into the riverbeds. As a result, heavy precipitation can cause flash floods with a sharp increase in the runoff of these rivers by 100–1000 times within a few hours [11,12].

Under normal conditions, river plumes in this region are clearly separated since their spatial scales do not exceed the distance between neighboring river mouths. However, spring melting of glaciers and spring-summer heavy rainfalls, which regularly occur in the considered region, cause a sharp increase in river runoff and synchronous flood formation along long stretches of the coast. During such flood periods, the areas of river plumes increase dramatically, and the spatial scales of many plumes begin to exceed the distances between neighboring river mouths, due to which neighboring river plumes start to merge and interact with each other. The most intense precipitation causes the formation of a continuous coastal strip of turbid water, which can be observed using satellite images. Since the end of the flood period, this stripe dissipates, and the areas of river plumes decrease to their mid-seasonal sizes.

In this work, we reconstruct the distribution of plumes in the study region during the low-discharge year (2020) and the high-discharge year (2021) using numerical modeling. Based on the obtained calculation results, we study the synoptic and inter-annual variability of small river plumes. We describe the effect of river discharge and wind variability on the area of the plumes and freshwater residence time within the plumes. Additionally, we study the velocity of the response of the plume dynamics at the beginning of a flooding event. Special attention is paid to the processes of formation and dissipation of the freshened alongshore stripe during spring or spring-summer freshets and rain-induced floods. Furthermore, we study the transport and accumulation of floating marine litter of riverine origin on the shoreline. Finally, we detect potential accumulation areas of floating marine litter in the study area, which depend on local external forcing conditions.

## 2. Materials and Methods

### 2.1. Satellite Observations

We studied the dynamics of river plumes in the northeastern part of the Black Sea using MODIS Terra/Aqua data. For the low-discharge year (2020), to capture moments before, during and after a large flooding events that occurred on 29 January 2020–10 February 2020 on the Sochi River and on 29 January 2020–14 February 2020 on the Mzymta River, satellite images were collected in January-February for the following dates: 20 January 2020, 4 February 2020, 10 February 2020, 18 February 2020. For the high-discharge year (2021), November-December period included several floods on the Sochi (24 November 2021–25 November 2021, 29 November 2021–1 December 2021) and Mzymta (29 November 2021–7 December 2021, 9 December 2021–10 December 2021) rivers, therefore we analyzed satellite images collected for the following dates: 25 November 2021, 3 December 2021, 7 December 2021, 13 December 2021. The MODIS Terra/Aqua products with 250 m spatial resolution were downloaded from the NASA repository of the satellite data (https://ladsweb.modaps.eosdis.nasa.gov/, accessed on 23 January 2023).

### 2.2. River Discharge in 2020–2021

In this study, we investigated seasonal and synoptic variability of plumes among the 9 (Figure 1) largest (but still small in terms of annual discharge rate) rivers of the northeastern part of the Black Sea for the low-discharge year (2020) and the high-discharge year (2021). We define large rivers as those with a catchment area from 100 to 1500 km$^2$ and an average annual discharge of 5–60 m$^3$/s: Psezuapse (14.05 m$^3$/s), Shakhe (28.13 m$^3$/s), Sochi (16.15 m$^3$/s), Mzymta (56.06 m$^3$/s). Small rivers have a catchment area from 15 to 100 km$^2$ and average annual discharge less than 5 m$^3$/s: Kuapse (0.75 m$^3$/s), Zapadny Dagomys (2.27 m$^3$/s), Matsesta (2.76 m$^3$/s), Khosta (5.05 m$^3$/s), Kudepsta (2.86 m$^3$/s).

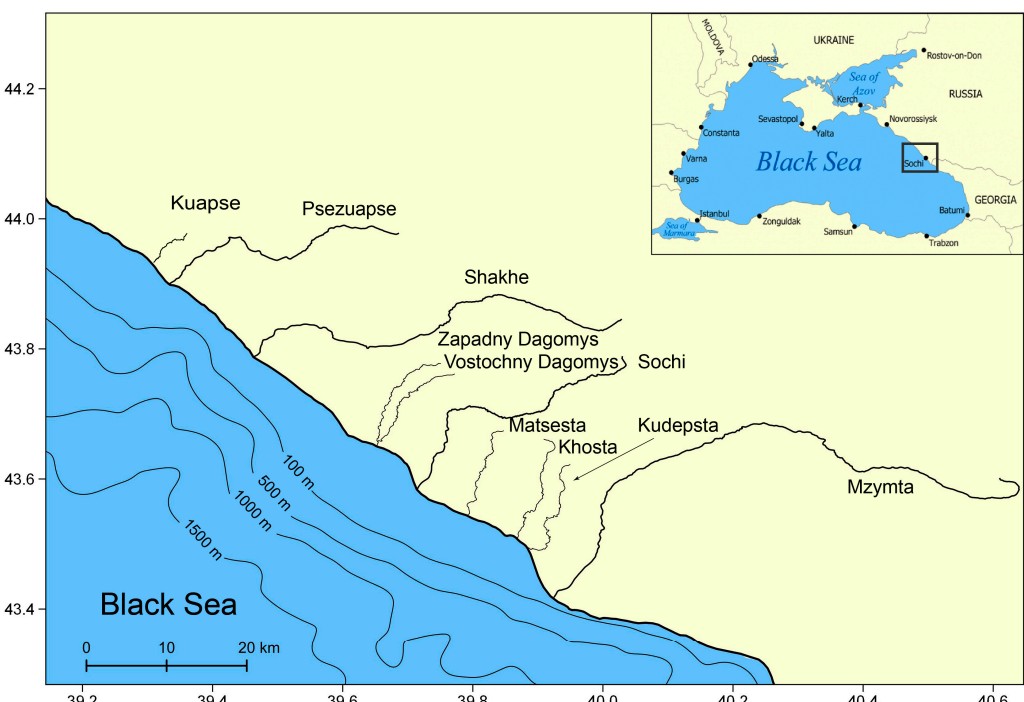

**Figure 1.** Study area, bathymetry, and locations of the rivers addressed in this study. The black box at the inset shows the location of the study area in the northeastern part of the Black Sea.

To study the seasonal and synoptic variability of river plumes in the considered region, we prepared the average daily freshwater discharge rates of the examined rivers for 2020–2021. For the Mzymta and Sochi rivers, they were made based on the data from the Russian Hydrometeorological Service (https://gmvo.skniivh.ru/, accessed on 23 January 2023) and 10-min water level data obtained from the Automated Flood Monitoring System of the Krasnodar Territory EMERCIT (http://www.emercit.com, accessed on 23 January 2023).

The Mzymta River is the largest river among the small rivers of the Caucasian coast of the Black Sea. Every year, it brings ~1.5 km$^3$ of water into the sea [10]. According to the obtained observational data (Table 1), the discharge of the Mzymta River in 2020 was 1.07 km$^3$ and 1.94 km$^3$ in 2021. For the Sochi River, the discharge amounted to 0.285 km$^3$ and 0.56 km$^3$ in 2020 and 2021, respectively. Thus, the mean annual discharge rates for these two rivers for the high-discharge year (2021) were 1.81–1.96 times more than for the low-discharge year (2020).

**Table 1.** Seasonal and annual freshwater discharges (km$^3$) for the Mzymta and Sochi rivers for 2020–2021 years and shares (%) of seasonal discharge from annual discharge. Abbreviation: w (winter), sp (spring), su (summer), au (autumn).

| River | 2020 | | | | | 2021 | | | | |
|---|---|---|---|---|---|---|---|---|---|---|
| | w | sp | su | au | year | w | sp | su | au | year |
| Mzymta (km$^3$) | 0.213 | 0.490 | 0.259 | 0.108 | 1.07 | 0.35 | 0.84 | 0.49 | 0.25 | 1.94 |
| Sochi (km$^3$) | 0.124 | 0.118 | 0.025 | 0.018 | 0.285 | 0.13 | 0.20 | 0.11 | 0.11 | 0.56 |
| Mzymta (%) | 20 | 46 | 24 | 10 | 100 | 18 | 44 | 25 | 13 | 100 |
| Sochi (%) | 43 | 41 | 9 | 6 | 100 | 24 | 36 | 20 | 19 | 100 |

Based on average daily discharges of the Mzymta and Sochi rivers (Figure 2) for two considered years, we additionally determined periods of floods (spring or spring-summer snowmelt freshets and rain-induced floods), their duration, maximum discharge peaks, and total discharge (Supplementary Materials—S1. Table S1).

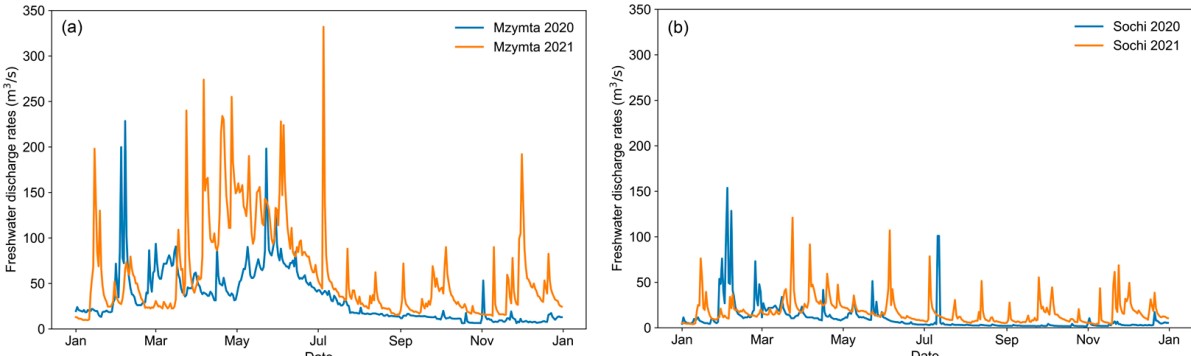

**Figure 2.** Freshwater discharge rates of the Mzymta (**a**) and Sochi (**b**) rivers during 2020–2021.

Floods were detected in the following way: their average daily discharge peak had to exceed earlier discharge by 2–2.5 times (with respect to a previous 1-day reference period) and exceed the long-term average monthly discharge (~30 years) in order to discard very low floods or discharge fluctuations. Snowmelt freshets and rain-induced floods were determined by taking into account the average daily air temperature and the amount of precipitation measured at the Sochi and Adler meteorological stations. Separation of precipitation into rain and snow was done according to the boundary value of air temperature of +2 °C [13]. The average daily discharges of the other seven rivers (Kuapse, Psezuapse, Shakhe, Zapadny Dagomys, Khosta, Matsesta, Kudepsta) were reconstructed based on the available climatic [10] and discharge data (https://gmvo.skniivh.ru/, accessed on 23 January 2023).

Summer, autumn, and the beginning of winter 2020 on the Black Sea coast were dry. Therefore, at the Sochi River (and other small rivers in the studied area), the low-discharge period lasted almost seven months: from 9 June 2020 to 10 January 2021. At the Mzymta River, a similar low-discharge period was established later, approximately from 6 August, after the end of the flood caused by snowmelt in the mid-mountain altitude zone. Because of the prolonged low-discharge period, the year of 2020 for the Sochi River became the next lowest in terms of annual discharge after 1986 during a 75-year-long observation period.

In 2021, due to snowfall and snowmelt in January and February, increased soil moisture, and frequent rains, floods on the rivers took place several times a month. The longest period between floods was only 21 days (from 14 June to 4 July), while the volume of summer discharge in 2021 exceeded the volume of the previous year by 1.65–2.26 times, and in autumn by 3.11–5.82 times for the Mzymta and Sochi rivers, respectively.

To study the necessity of using hourly river discharges instead of daily data in the numerical calculation of the northeastern part of the Black Sea, we additionally obtained hourly freshwater discharges of nine rivers for two severe July flash floods in 2021 (4–6 and 22–24 July). For most of the rivers (Psezuapse, Shakhe, Sochi, Mzymta, Kuapse, Khosta), the discharges were provided by the SCHME BAS (Special Center on Hydrometeorology and Monitoring of Environment of the Black and Azov Seas, https://www.pogodasochi.ru/, accessed on 23 January 2023). Discharge rates for other rivers (Dagomys, Matsesta, and Kudepsta) were evaluated using the KW-GIUH model. KW-GIUH (Kinematic Wave-Geomorphologic Instantaneous Unit Hydrograph) is an event-based rainfall-runoff model, which is an efficient tool for reconstructing the hydrological response of a river catchment on intense precipitation. The model uses geomorphological information about the characteristics of lengths and slopes of river sub-catchments and channels. The travel time of slope and channel runoff for each sub-basin is estimated using formulas derived from kinematic wave theory [14–16]. Geomorphological characteristics (their lengths and slopes) were obtained using MERIT DEM data (http://hydro.iis.u-tokyo.ac.jp/~yamadai/MERIT_DEM/, accessed on 23 January 2023) [17] and ArcGIS tools. The channel width at the river mouths was determined using satellite images. The slope and channel roughness coefficients were calibrated for previous flood events on 25 June 2015 and 24–25 October 2018 [18]. Initial

data for the model setup were hourly precipitation measured at automatic weather stations of SCHME BAS.

### 2.3. INMOM

INMOM (Institute of Numerical Mathematics Ocean Model) Eulerian model of marine circulation was used to calculate surface salinity and current fields in this study. INMOM is a three-dimensional σ-model based on primitive equations in hydrostatic and Boussinesq approximations [19–21]. The regional version of INMOM has previously been used in several studies on coastal circulation, river plumes, and pollutant transport [18,22,23]. A brief description of the model and input data is given below and more detailed information is presented in the Supplementary Materials—S2.

A high-resolution regional version of INMOM with a non-uniform horizontal grid covered the Black Sea basin, excluding the Azov and Marmara seas. The model horizontal grid contained 642 × 715 points in longitude and latitude, respectively. To implement a non-uniform horizontal grid in the model, a spherical coordinate system was used with one of the poles located in the study area (40.205° E, 43.84° N). The horizontal spatial resolution was from ~200 m near the pole to ~4.5 km in the southeastern part of the Black Sea (Figure 3). In the vertical direction 20 σ-levels, unevenly distributed over depth, were set with a concentration near the surface to achieve higher resolution.

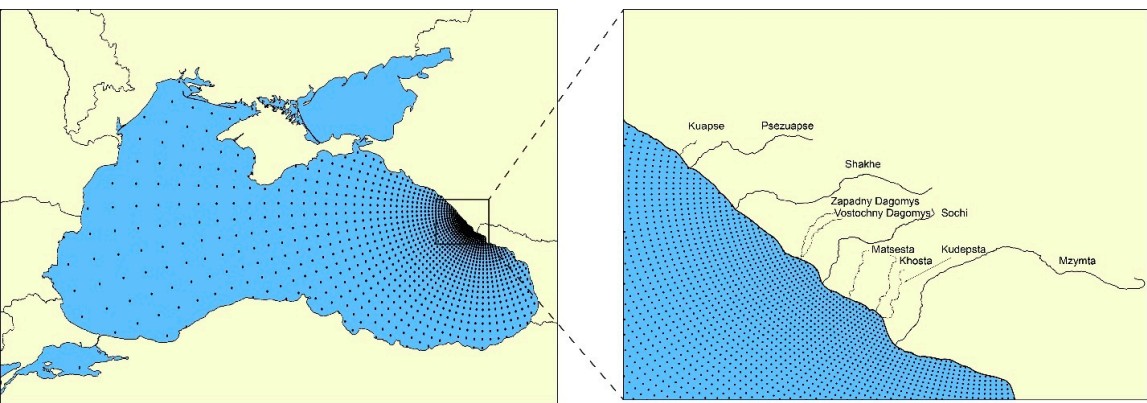

**Figure 3.** INMOM model domain at the Black Sea (every 15th grid point is shown) (**left panel**) and at the study area (every 5th grid point is shown) with an indication of the location of the considered rivers (**right panel**).

Model bathymetry was set using GEBCO data with a spatial resolution of 15″ (www. gebco.net, accessed on 23 January 2023). Atmospheric forcing, including turbulent heat, salt, and momentum fluxes, was calculated based on the data on shortwave and longwave radiation, air temperature, precipitation, relative humidity, sea level pressure and wind obtained from the atmospheric regional non-hydrostatic model WRF (Weather Research and Forecasting Model) [24]. WRF model was developed by NCEP (National Center for Environmental Prediction) and NCAR (National Center for Atmospheric Research) in the USA and now, even more institutions are involved in its development. The Era-Interium (ECMWF Re-Analysis) reanalysis was used for the model boundary and initial conditions. The data on the underlying surface were taken from the MODIS (Moderate Resolution Imaging Spectroradiometer) data. Detailed description of the model is presented in the following work [24]. WRF model was implemented in the SOI (N.N. Zubov State Oceanographic Institute, Roshydromet) with a spatial resolution of 10 km and a time step of 1 h. Water transport through the Kerch and Bosphorus straits connecting the Black Sea with the Sea of Azov and the Sea of Marmara, as well as the flow of large Black Sea rivers, including the Danube, the Dniester, the Dnieper, the Kodori, the Rioni, the Inguri, the Yeshilirmak, the Kyzylirmak, and the Sakarya were set using available climatic data [10]. For nine main rivers of the examined area, described in Section 2.2, the reconstructed

average daily data was used, except for additional calculations, when hourly data was used for July rain-induced flash floods (4–6 and 22–24 July 2021). Three-dimensional monthly mean climatic thermohaline fields for the Black Sea with a horizontal resolution of $0.1° \times 0.0625°$ and with 36 vertical z-levels from 0 to 2150 m were provided by MHI RAS (Marine Hydrophysical Institute of the Russian Academy of Sciences) [25]. These data were used as the initial state of the model. All input data were interpolated into the model grid. The model time step was 90 s, and the output data resolution was one hour.

### 2.4. OpenDrift

OpenDrift was used to calculate the residence time of freshwater in the Mzymta and Sochi plumes and the areas where riverine Lagrangian particles were captured at the shoreline. OpenDrift is an open-source software package (https://github.com/OpenDrift/opendrift/, accessed on 23 January 2023) developed at the Norwegian Meteorological Institute for modeling the transport of Lagrangian particles [26]. A distinctive feature of this complex is its modular structure and versatility in the setting of external forcing. The complex is based on the transfer of Lagrangian particles, using which many modules have been implemented: an oil spill modeling module, a module for stochastic modeling of search and rescue operations, a biological module for calculating the ichthyoplankton transport and an assessment of the stocks of pelagic eggs and larvae in the water column. In addition, this complex can be used to model the transport of floating marine litter and microplastics [27]. More description is presented in the Supplementary Materials—S3. In this study, we used hydrometeorological characteristics (fields of near-water wind and current velocities) calculated by WRF and INMOM models [19–21,24] as initial data for calculations of plumes features. The Stokes drift velocity was set according to the available data of the European CMEMS service: for the period from 1 January 2020 to 1 July 2020—according to the global analysis and forecast of wave characteristics (GLOBAL_ANALYSIS_FORECAST_WAV_001_027), and from 1 July 2020 to 1 January 2022—according to the analysis and forecast of wave characteristics for the Black Sea (BLKSEA_ANALYSISFORECAST_WAV_007_003). Surface current velocities calculated by INMOM model with a time resolution of 1 h were interpolated into a geographic coordinate system with cell sizes of 250 m × 250 m. We considered that the Lagrangian particles moved under the influence of currents, the Stokes drift speed, and 2% of wind speed. Every hour in grid cells of 250 m × 250 m in the vicinity of small river mouths (Psezuapse, Shakhe, Sochi, Mzymta, Kuapse, Zapadny Dagomys, Khosta, Matsesta, Kudepsta), a certain amount of floating Lagrangian particles was generated. The number of particles was proportional to river discharges. The calculation periods corresponded to all floods for 2020 and 2021 (Supplementary Materials—S1. Table S1) taking into account daily data on river discharges at specific periods. For 2021, additional calculations of July floods (4–6 and 22–24 July) were carried out considering hourly river discharges to find the sufficient time resolution of discharge rates used for such studies. In the calculations, it was assumed that when particle collided with the shore it was trapped at the shore and its position did not change after that.

### 2.5. Numerical Models Coupling

The way models from Sections 2.2–2.4 (KW-GUIH, WRF, INMOM, OpenDrift) interact with each other presented on the flowchart (Figure 4). To reproduce hydrological characteristics of the studied area using INMOM we set atmospheric characteristics from WRF and river discharge data from KW-GIUH. We use calculated current velocities and salinity fields by the INMOM and wind velocities by the WRF as input data in OpenDrift to define trajectories of particles.

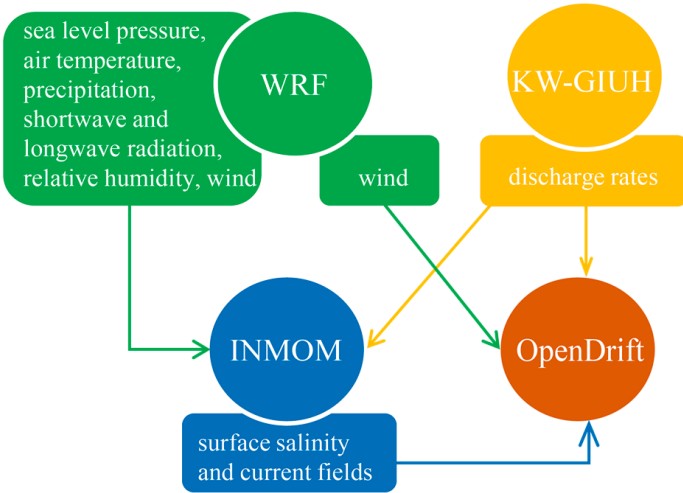

**Figure 4.** Flowchart of the models interaction.

### 2.6. Plume Area Calculations

To calculate the plume areas of the Mzymta and the Sochi rivers, KMeans cluster analysis from Scikit-learn (Sklearn) library for machine learning in Python was used. Using this technique, we checked if points near river mouths were inside river plumes based on the fields of surface salinity. A plume was an area with salinity that did not exceed 16.5 psu. Such reference value of salinity made it possible to identify plumes during the period of low discharge, when plumes formed in the vicinity of river mouth and quickly mixed with sea water of higher salinity. So, if salinity in a given point was less than the threshold value of 16.5 psu, then this point was classified as a member of the specific river plume cluster. It was necessary to set the total number of clusters as the initial data for this method. Due to the problem of determining this total number, an automatic clustering method was used but its accuracy was not always sufficient. Although river mouths in the northeastern part of the Black Sea were located at some distance from each other, it was already noted in the previous work [18] that an area of plumes from small rivers increases significantly under flood conditions. The interaction of river plumes with each other significantly affects their structure and dynamics. As a result, neighboring plumes merge, which leads to the formation of a continuous stripe of freshened water with low salinity and high turbidity, the total length of which can exceed 200 km. Consequently, there were cases when it was difficult to distinguish between points belonging to neighboring river plumes using automatic classification method, so manual adjustments were made occasionally.

### 3. Results

### 3.1. Satellite Observations

We collected and analyzed the most representative available satellite images without cloud coverage for the low-discharge year (2020) on 20 January and 4, 10, 18 February and for the high-discharge year (2021) on 25 November and 3, 7, 13 December (Figure 5). Such coverage of satellite data provided an opportunity to examine the dynamics of river plumes formed in the study area before, during, and after long and intense floods, noted from 29 January until 10 February 2020 for the Sochi River and from 29 January until 14 February 2020 for the Mzymta River (Supplementary Materials—S1. Table S1). For 2021, we examined period in November-December with not so long and intense like in 2020, but numerous floods.

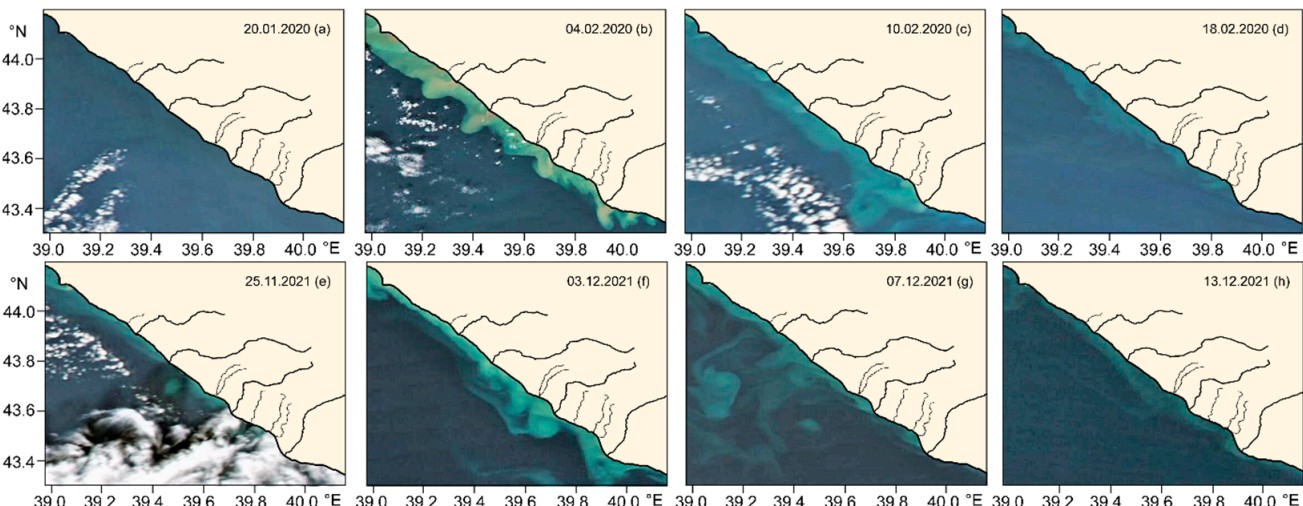

**Figure 5.** MODIS satellite images (visible band) of the study area demonstrating the considered river plumes on 20 January (**a**), 4 February (**b**), 10 February (**c**), 18 February (**d**) 2020 and 11 November (**e**), 3 December (**f**), 7 December (**g**), 13 December (**h**) 2021.

Before the flooding event (Figure 5a), daily discharges for the Sochi and Mzymta rivers were 4.0 and 16.1 $m^3/s$, respectively, which corresponded to baseflow and was about 3–4 times less than the average monthly values of 18.5 and 40.1 $m^3/s$, therefore, on satellite images, plumes from rivers were invisible. However, during the flooding event period on February 4 (Figure 5b), the influence of strong E-SE wind (more than 12.5 m/s) and high daily discharges (191 and 208 $m^3/s$ for the Sochi and the Mzymta rivers respectively) was clearly visible. They contributed to the movement of plumes in the NW direction and formed a clearly distinguishable wide strip of fresh water along the coast. Under the influence of frequent wind changes, first counterclockwise and then clockwise, on the date of the image, plumes spread to the deeper part of the sea. On February 10 (Figure 5c), a frequent change in the wind prevailed, firstly counterclockwise and then clockwise, so plumes, for the most part, also propagated to the deeper part of the sea. However, due to reduced values of river discharges (21.3 and 51.0 $m^3/s$ for the Sochi and the Mzymta rivers respectively) and wind amplitude (less than 7.5 m/s), plumes were more diffuse. After the flooding event on February 18 (Figure 5d), discharges (10.3 and 24.8 $m^3/s$ for the Sochi and the Mzymta rivers, respectively) and wind amplitude decreased even more but still remained large, so the alongshore strip was no longer observed but individual plumes were present.

From 24 to 25 November , there was a relatively strong flooding event on the Sochi River with discharges of 68.7 and 25.7 $m^3/s$, while on the Mzymta River there was also a one-day flooding event (24 November ) with a daily discharge rate of 78 $m^3/s$, the wind changed from SE to NE, but did not exceed 2.5 m/s. Hence, plumes were visible on the satellite image, but those that were visible concentrated near the mouths (Figure 5e). 3 December (Figure 5f) was the end of the flooding event on the Sochi River (form 29 November until 3 December), the discharge on the last day was 27 $m^3/s$. For the Mzymta River it, was the middle of the flood (from 29 November until 7 December), the discharge on 3 December reached 88.1 $m^3/s$. During the first three days of the flooding event, the discharges of the Mzymta River reached 192 $m^3/s$, and of the Sochi River, 33.3 $m^3/s$; wind speed reached 15 m/s, which significantly affected the plume pattern on the analyzed date. The freshened alongshore strip was visible, but due to strong wind changes clockwise and counterclockwise from SE, after 30 November, distribution to the deeper part of the sea and concentration of plumes near mouths could be traced. On 7 December, the flooding event on the Mzymta River still continued, but the discharge also decreased (51 $m^3/s$) and the SE wind up to 10 m/s began to predominate, so the freshened strip that spread into the

sea began to spread to the NE and finally dissipated. Until 13 December (Figure 5g), there was another flooding event on the Mzymta River with discharge rates (45.4 and 49.9 m³/s), the wind was changing from NE to SE, amplitude values were about 5 m/s, so the strip was still visible on the images, but it was gradually dissipating.

### 3.2. Seasonal Freshwater Discharges in 2020–2021

A river plume is a freshened water mass, which is formed because of the river and seawater mixing on a daily or synoptic time scale. The salinity of the plume is lower than the salinity of the surrounding sea. Therefore, salinity is an important characteristic of the marine environment and the main indicator for determining the boundary between the river plume and seawater. Since river discharge is subject to seasonal and inter-annual variability, the salinity distribution also changes depending on the season and year.

In the considered region, all rivers are mainly mountainous and are characterized by annual spring or spring-summer snowmelt freshets [10]. Regardless of the season, a significant but short-term rise in river water level can occur due to rain-induced floods. Both spring and spring-summer snowmelt freshets with superimposed high rain peaks and heavy rain-induced flash floods can lead to inundation. Outliers in discharges of the Mzymta and Sochi rivers correspond to rain-induced flash floods (Figure 2).

We initially analyzed daily-mean river discharges for 2020–2021 (Figure 2) to study seasonal and inter-annual variability of the Mzymta and the Sochi river plumes. The analysis included freshwater discharges for the Mzymta and the Sochi rivers by season and per year (Table 1). We also considered the ratio of seasonal and annual discharges for two years (high and low waters). Next, we investigated monthly-mean and seasonal fields of surface salinity in the northeastern coast, obtained from the hourly data from INMOM model calculations, as well as seasonal wind roses in the mouths of the Mzymta and the Sochi rivers (Supplementary Materials—S1. Figure S1). Seasonal surface salinity figures and wind roses near the Mzymta River mouth for the high-discharge year (2021) (Figure 6) were similar to those for the low-discharge year (2020), but more indicative) (Figure 7).

It was previously noted that the ratio of the Mzymta and the Sochi river discharges between the high-discharge year (2021) and the low-discharge year (2020) was more than 1.7 times and mainly depended on the number of floods that occurred. In 2020 and 2021, there were 2 and 13 floods on the Mzymta River, 4 and 16 floods on the Sochi River, respectively. In 2020, there were 2 winter rain-induced floods on the Mzymta River and 1 on the Sochi River, and in 2021 for each of the rivers there were about 3 times more floods (7 for Mzymta and 3 for Sochi). However, in the winter season, the discharge of the Mzymta River differs by 1.52 times, and for the Sochi River, the total discharge for two years is the same. This means that in addition to the number of floods during the year, it is necessary to analyze their duration and intensity.

In the winter season of 2020, the discharge of the Mzymta and Sochi rivers amounted to about 20% (0.213 km³) and 43% (0.124 km³) of their annual discharge (1.07 and 0.285 km³) (Table 1), mainly due to the long and intense rain-induced floods, noted on two rivers in January and February. The floods began simultaneously on January 29, lasted 17 and 13 days, with maximum discharge peaks reaching 228 m³/s and 153.7 m³/s for the Mzymta and the Sochi rivers respectively. In total, during the flooding period, the runoff of the Mzymta and Sochi rivers amounted to half (0.109 and 0.071 km³) of the total discharge for the winter season.

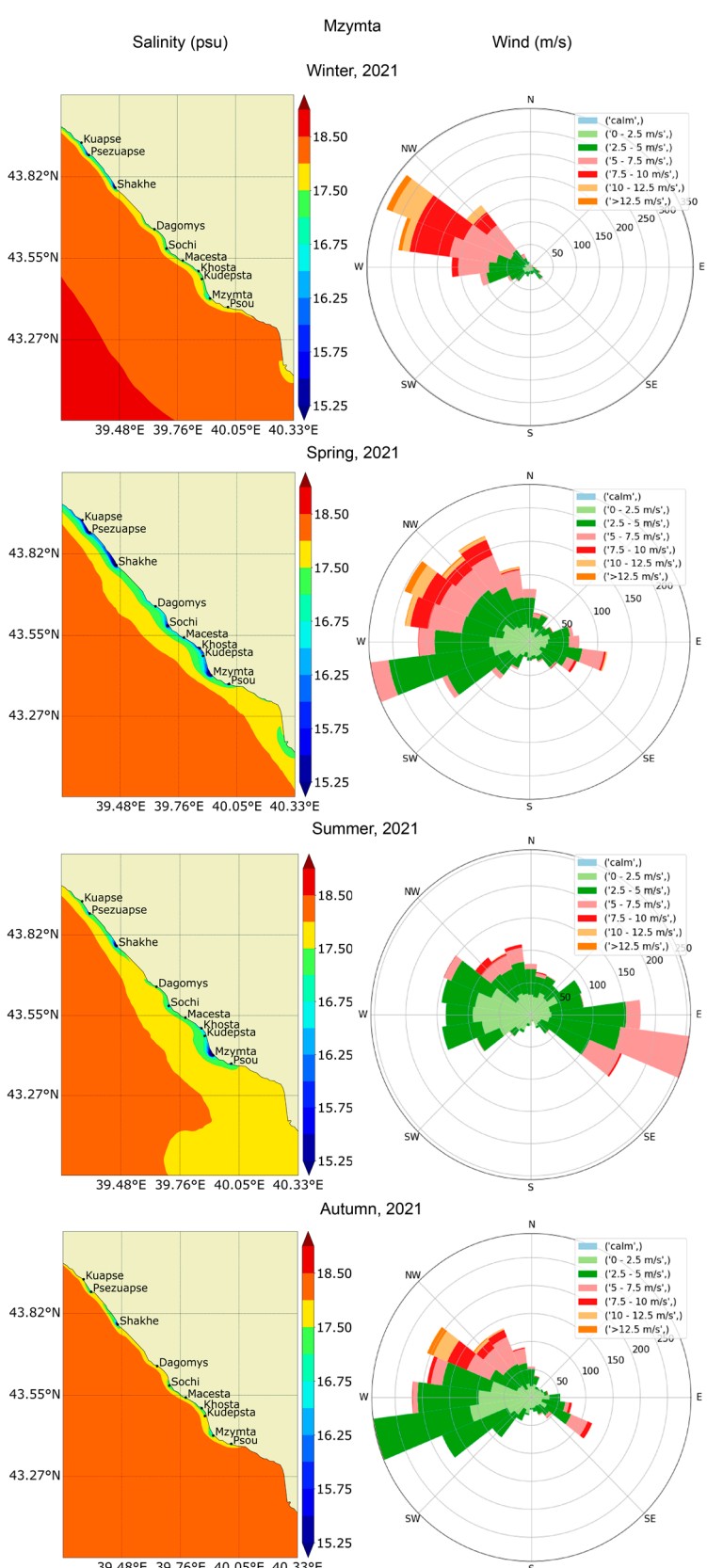

**Figure 6.** Seasonal fields of surface salinity (psu) according to the INMOM model (**left**) in the northeastern coast of the Black Sea and wind roses (m/s) according to the WRF model (**right**) near the Mzymta River mouth for the high-discharge year of 2021.

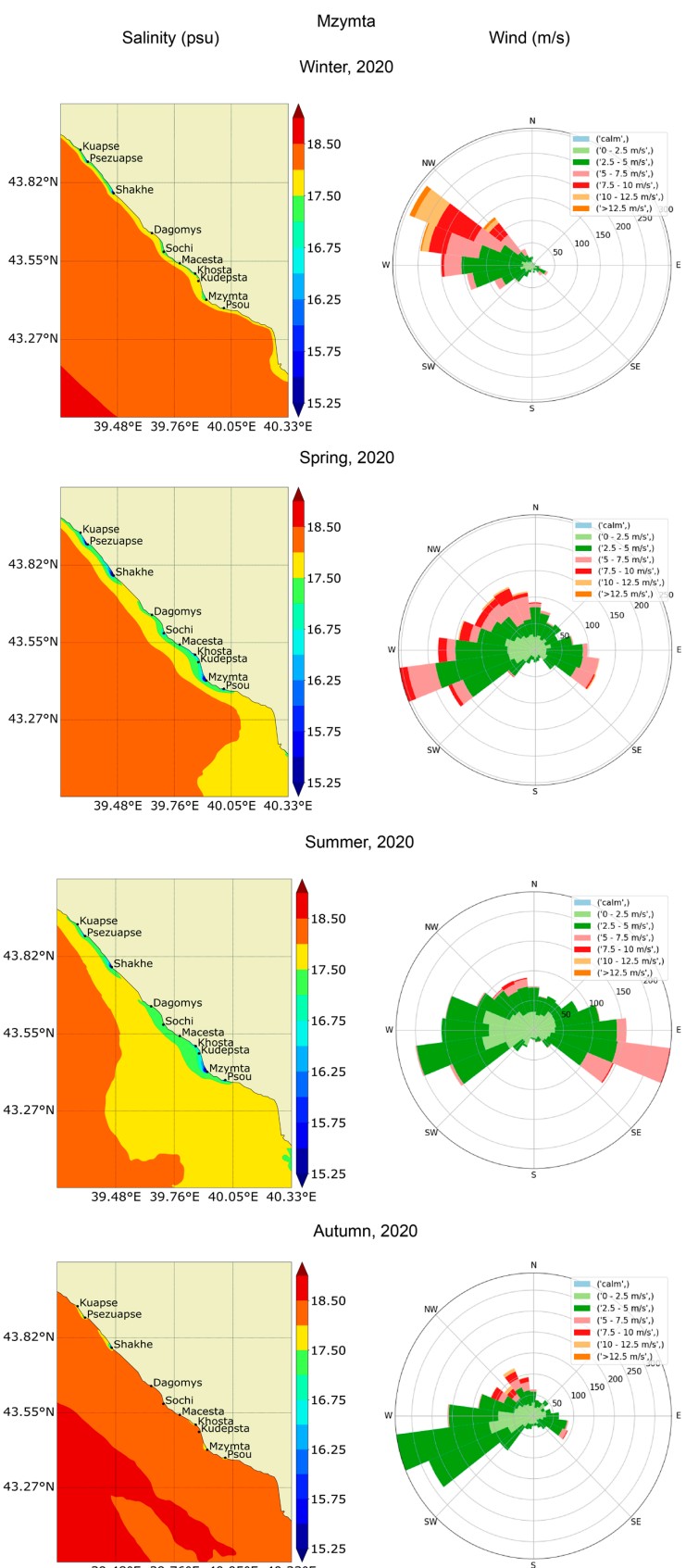

**Figure 7.** Seasonal fields of surface salinity (psu) according to the INMOM model (**left**) in the northeastern coast of the Black Sea and wind roses (m/s) according to the WRF model (**right**) near the Mzymta River mouth for the low-discharge year of 2020.

For the winter of the high-discharge year (2021), the discharges of the Mzymta and the Sochi rivers were 18% (0.35 km$^3$) and 23% (0.13 km$^3$) of their annual runoff (1.94 and 0.56 km$^3$). In 2021, long (from 12 to14 days for the Mzymta River and from 11 to 13 days for the Sochi River) and joint rain-induced floods for two rivers were also observed in January (10 January/11 January–21 January) and February (5 February–17 February/ 18 February). However, maximum flood peaks did not exceed 137.9 m$^3$/s for the Mzymta and 76.2 m$^3$/s for the Sochi. Discharges during these two floods were only about 40% (0.149 and 0.051 km$^3$) of the total winter runoff. The analysis of wind roses showed that E-SE winds prevailed in winter, with a magnitude sometimes exceeded 10 m/s, and weaker E-NE winds (up to 7.5 m/s) occured. Such wind directions mainly contributed to the movement of currents in the NW direction; therefore, a narrow strip of fresh water along the coast was visible in the salinity fields. Since wind roses were similar between years, the difference in surface salinity fields and greater freshening of seawater along the coast in 2021 was mainly affected by the previously analyzed difference in river discharges.

In contrast to the winter season, discharge ratios for spring and summer between two years did not differ so much from the annual ratios (1.81–1.96 km$^3$), so the difference between the surface salinity patterns was more noticeable. Spring-summer snowmelt freshets of the Mzymta River (24 February–31 July 2020 (159 days, 0.75 km$^3$) and 16.03–05.08 2021 (143 days, 1.241 km$^3$)) and less intense and short-term spring snowmelt freshets of the Sochi River (24 February–22 March 2020 (28 days, 0.059 km$^3$) and 1 April–5 May 2021 (35 days, 0.094 km$^3$)) made a significant contribution to the surface salinity patterns of the spring-summer seasons. As in winter, freshening near river mouths was affected by the river flow, and freshening along the coast or in the deeper sea was influenced by the wind direction. In spring, SE-SE and W-NW winds were added to general SE-E-NE wind directions, which contributed to plume stretching along the coast, indicating a wind change. When the wind changed clockwise, the plumes concentrated in the vicinity of river mouths. In 2021, the SE-E-NE winds were more frequent and intense than in 2020 (up to 12.5 m/s), which contributed to the elongation and merging of freshened seawaters along the coast. In summer, the NE-E-SE-S winds weakened to 5.0 m/s, W-NW winds intensified up to 10 m/s, and river discharge decreased compared with the spring season, which affected the final patterns of seasonal salinity.

In autumn, the discharge ratios between the two years differed significantly, affecting the resulting salinity patterns. The prevailing directions of the wind roses of the autumn season were like the spring wind roses, and the main difference was a smaller wind magnitude.

The analysis showed that the general patterns of seasonal fields of surface salinity for the low-discharge (2020) and the high-discharge (2021) years were affected by the period of occurrence, the number, the duration and intensity floods, namely the difference in river discharge values and the prevailing values of wind direction and magnitude.

### 3.3. Seasonal Variability of Plume Areas in 2020–2021

In this study, we analyzed total and average plume areas per season, obtained using daily area data, for a quantitative assessment of the Mzymta and Sochi river plumes. As it was mentioned in Section 2.6, a plume was an area with salinity that did not exceed 16.5 psu. The variability of plume areas depended on many factors (river discharge, wind fields, hydrological features of the water area, coastal shape, etc.). According to the results of seasonal variability (Figure 8) of total plume areas, the influence of intense rain-induced floods (winter), spring or spring-summer snowmelt freshets with overlapping rain-induced floods (spring-summer season), and weak rain-induced floods (autumn) was clearly visible.

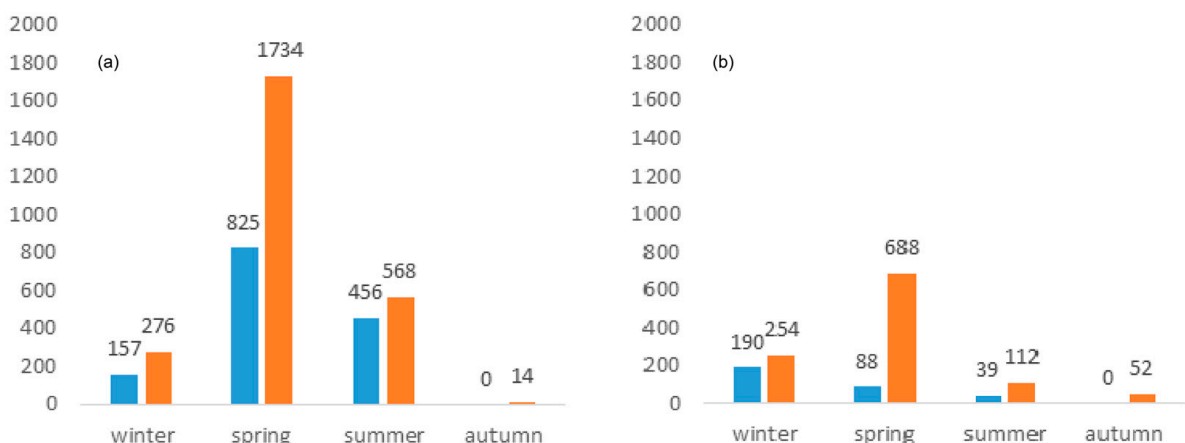

**Figure 8.** Seasonal variability of total plume areas (km$^2$) from the Mzymta (**a**) and Sochi (**b**) rivers in 2020–2021.

However, in addition to the influence of the river discharge, spatial-temporal structure of the wind field significantly affected plume structure. During the winter seasons in 2020 and 2021, wind fields near mouths of the Mzymta and the Sochi rivers were similar. When SE-E-NE wind prevailed (Figures 6 and 7, Supplementary Materials—S1. Figure S1) with amplitude up to 10 m/s, it turned out that the river discharge values (Table 1) affected seasonal areas of plumes. With a decrease in the wind amplitude and change in the general wind direction (Figures 6 and 7, Supplementary Materials—S1. Figure S1) in spring-summer, the dependence of the seasonal plume area on the river discharge decreased. It could be assumed that with frequent changes in the wind direction, the influence of the wind on the plume area increased. A similar situation was typical for autumn, when the SE-E-NE wind returned, but with a continuing change in the wind. Under the influence of frequent wind changes, the Mzymta and the Sochi plumes were more often concentrated near river mouths or, on the contrary, spread to the deeper part of the sea, which affected plume areas. An analysis of daily-mean plume areas revealed that the maximum value of the plume area of the Mzymta River was a little more than 70 km$^2$ in May 2020 and a little less than 70 km$^2$ in April 2021, and for the Sochi River—about 20 km$^2$ in February 2020, and 30 km$^2$ in April 2021. It turned out that high values of river discharges do not guarantee the largest area of the river plume. In 2020, for the Mzymta River, the largest discharge was observed in July, and the largest area was in May.

Thus, due to the inter-annual and intra-annual variability of river discharges and the variability of the spatial and temporal structure of the wind field, seasonal analysis was not enough to establish an unambiguous effect of these parameters on plume areas and it was necessary to study synoptic variability by analyzing each of the flood events separately.

### 3.4. Synoptic Variability of Rain-Induced Floods in 2020–2021

We analyzed all previously identified floods (Supplementary Materials—S1. Table S1) for the Mzymta River (2 in 2020; 13 in 2021) and the Sochi River (4 in 2020; 16 in 2021) to study the synoptic variability of plumes in high-discharge (2021) and low-discharge (2020) conditions. We consider periods of floods, their duration, maximum discharge peaks, total discharge, the number of Lagrangian particles left in the sea and washed ashore, the prevailing wind direction, the frequency of its change, and the maximum wind amplitude at the mouths of rivers, as well as freshwater residence time within plumes for each of the floods (Figure 9).

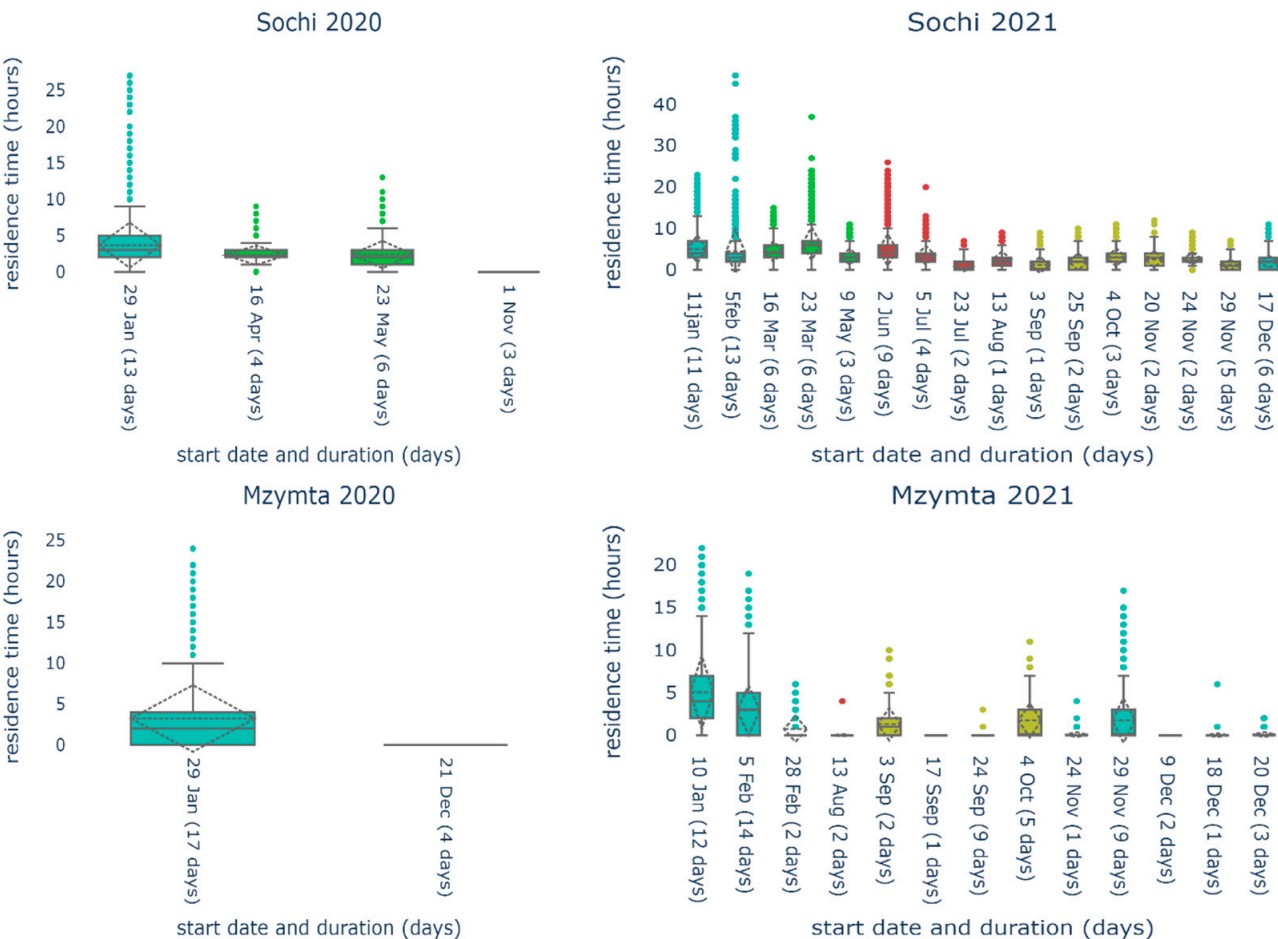

**Figure 9.** The freshwater residence time within the plumes (hours) for floods for low-discharge year (2020) (**left**) and the high-discharge year (2021) (**right**) on the Mzymta (**bottom**) and the Sochi (**top**) rivers. Box plots show the median, upper and lower quartiles, minimum and maximum values, and all outliers in the dataset. The dotted line shows the average value, and the color indicates the seasons of flood occurrence (turquoise—winter, green—spring, red—summer, yellow—autumn).

For rain-induced flood periods, we analyzed hourly wind data near river mouths 12 h before the onset of the flood itself. Note that the previous wind and seawater freshening near river mouths caused by an increase in river discharge significantly affected the further distribution of plumes. Based on obtained simulation results, the wind effect was in good agreement with the main results of the works [28,29], where the influence of different wind-forcing conditions (downwelling, onshore, upwelling, offshore, low wind) on the distribution of the Mzymta River plume was investigated. It was noted, that onshore winds caused upstream accumulation of small river plumes and other winds caused their downstream and offshore spreading. In the case of the Sochi River, the effect was similar; only it was necessary to take into account the shape of the coast in the vicinity of the mouths. In particular, the mouth of the Mzymta River is extended into the sea further than the Sochi River mouth, which affects the pollution of the coastal and marine zones under the influence of different winds. In addition, it was necessary to consider the change in wind direction. In the case of a clockwise wind change from SE to NW, the particles were thrown into the shore near the mouth. With a counterclockwise wind change, the river waters spread to the offshore areas, where, as a result of mixing with seawater, lower salinity values were observed compared to the surrounding waters. It should also be taken into account that during and after the low-discharge period, the rain-induced flood must be intense enough to freshen the water to the chosen value of 16.5 psu, which in this study

was considered the boundary of the plume. Therefore, the freshwater residence time in plumes strongly depended on wind duration, intensity, and direction during the flood period. Nevertheless, even under the most favorable wind conditions, this value rarely exceeded 6 h.

The number of Lagrangian particles washed ashore with river waters during flood periods directly depended on the duration and intensity of these floods, wind dynamics (Supplementary Materials—S1. Table S2), and Stokes drift. With E-SE-S wind direction and sufficient wind magnitude (~10 m/s), particles were mostly washed ashore northwest of the river mouths. With a W-NW-N wind, they, on the contrary, washed ashore southeast of the river mouths.

There were several factors favorable for the formation of a stripe along the northeastern coast: SE wind with an amplitude of about 10 m/s without a sharp change in direction during the day, high flood peaks (more than 30 m$^3$/s, except for the low-discharge period) or a large total river discharge. Furthermore, when the wind changed counterclockwise, all particles were mostly advected to the sea, and when the wind changed clockwise, the particles propagated in the SE direction.

As part of the experiment, Lagrangian particles were released from the nine studied rivers along the northeastern coast of the Black Sea during severe rain-induced flash floods in July 2021. To test if the setting of the hourly-mean discharges of rivers was required in the framework of such studies, we determined areas of particle accumulation on the shore. Under the wind and coastal current influence, the particles gradually spread along the entire coast, mainly in the northwest direction, but some particles also moved in the southeast direction. Some particles penetrated as far as the Gelendzhik Bay in the northern part of the model domain, and some of the particles were carried into the deep part of the sea. But in most cases, particles were carried ashore. So, they could be found along the entire northeastern coast of the Black Sea. Moreover, there was practically no difference in the distribution of particles along the coast, provided that different values of river discharges (daily-mean or hourly-mean) were set. According to the obtained results, it can be assumed that within this study, hourly-mean discharges were not necessary (Table 2).

**Table 2.** The number of Lagrangian particles washed ashore during the July rain-induced flash floods in 2021.

|  | Total | Washed Ashore |
|---|---|---|
| 2021, daily-mean discharge | 283,608 | 201,893 |
| 2021, hourly-mean discharge | 291,790 | 194,654 |

Compared to the previous work with the reproduction of the rain-induced flash flood in October 2018 [18], there were no significant differences in the propagation of Lagrangian particles. Lagrangian particles marked the boundaries of individual river plumes and specified their drift trajectories from the river mouth, and the degree of plumes mixing with each other. Initially, plumes interacted as separate structures with clear boundaries, but later these boundaries dissipated, and plumes from the considered small rivers merged into a single freshened narrow stripe, as shown in previous work [18], or into several stripes. As a result of advection, plumes were stretched along the coast and mixed with. Only the boundary between coastal freshened waters and the salty sea water remained.

## 4. Discussion and Conclusions

This work investigates the influence of wind and river discharge conditions on variability of small river plumes located in the northeastern part of the Black Sea using numerical modeling techniques. We focus on 9 small rivers in the study area: Psezuapse, Shakhe, Sochi and Mzymta rivers with an average annual discharge within the range 5–60 m$^3$/s; Kuapse, Zapadny Dagomys, Matsesta, Khosta and Kudepsta rivers with an average annual discharge < 5 m$^3$/s. These river plumes have local influence on the coastal areas, except the

periods of spring-summer freshets and rain-induced floods. During these periods, runoff of small rivers significantly increases by 100 or even more times within a few hours greatly expanding the plume areas. This feature determines the importance of small river plumes for local coastal processes.

In this study, we investigated synoptic and seasonal variability of these small river plumes. For this purpose, we simulated river plumes during the low-discharge year (2020) and the high-discharge year (2021) which have significant differences in annual river discharge volume and temporal distribution of flooding periods. According to reconstructed monthly-mean and seasonal surface salinity fields, small river plumes were almost constantly present in the considered region during both years. The largest plume areas were observed, first, during spring or spring-summer snowmelt freshets and, second, after intense rain-induced floods. Despite the difference in total river discharges during the low-discharge or high-discharge years, the intensity of water freshening in coastal areas depends more on the duration of the flood events and the related volumes of river discharges than wind magnitude and direction.

Seasonal variability of the plume areas depends mainly on river discharge rate and wind forcing. Synoptic, daily, and hourly variability of wind forcing governed significant changes in plume structures with a time lag of several hours, which is much smaller than previously reported for larger river plumes [30,31]. Similarly, freshwater residence time in the plumes strongly depends on duration, intensity, and direction of wind forcing during flood periods. Nevertheless in all cases it does not exceed 6 h. Considering changes in wind direction, the Mzymta and Sochi plumes were mainly concentrated near river mouths (in case of clockwise wind change from SE to NW). The opposite situation, i.e., offshore spreading of river plumes to the deep part of the sea (in case of counterclockwise wind change from SE to NW) was observed relatively rarely. High river discharge often did not result in the largest area of the river plumes, because the effect of high discharge was reduced by synoptic and seasonal patterns of atmospheric circulation.

In order to study the fate of river-borne floating litter, we carried out numerical experiments with Lagrangian particles, which were released from river mouths during rain-induced flash floods in July 2021. We revealed potential areas of particle accumulation at the shoreline. Most of the particles penetrated in the northwest direction along the shore according to the Coriolis force and the general circulation of the Black Sea. Additionally, flash floods in July 2021 were studied, taking into account the hourly-mean and daily-mean discharges of small rivers. It was shown that usage of daily-mean values of river discharges was sufficient for the correct reproduction of spreading of small river plumes and the related transport of river-borne floating litter.

Despite the importance of the obtained results for better understanding of spreading of river plumes in the northeastern part of the Black Sea, this work has broader applicability for studying river plumes located in other coastal areas of the World Ocean. First, we demonstrate that seasonal and synoptic features of local atmospheric circulation could strongly modify the relation between river plume characteristics and river discharge rate. As a result, there should be no direct dependence between river discharge rate and river plume area at least for small river plumes.

Second, we reveal that average area of a small river plume is determined not only by total river discharge volume during a certain time period, but also strongly depends on temporal distribution of discharge. In particular, homogenously distributed river discharge during a week/month/year will result in smaller average plume area as compared to river discharge with alternating flash flooding and drought conditions. This result highlights the importance of correct registration of flash floods and their representing in numerical modeling of river plumes. Simulations of small plumes, which neglect flash floods due to lack of related discharge data, could provide incorrect results.

Third, this work contributes to the important problem of temporal resolution of external forcing conditions for modeling of small river plumes. We demonstrate that the temporal resolution of wind data should be not greater than several hours due to very small response time of small plumes on wind variability. River discharge data, on the opposite, could be used with coarser (daily) resolution, because even small river plume slackens hourly variability of river discharge. Finally, in this study we report very small residence time of freshwater within small river plumes, which is <6 h. This temporal resolution should be considered as the time scale for transport of dissolved pollutants from river mouth to sea.

**Supplementary Materials:** The following supporting information can be downloaded at: https://www.mdpi.com/article/10.3390/w15040721/s1, Figure S1: Seasonal fields (a–d) of surface salinity according to the INMOM model results in the northeastern part of the Black Sea and wind roses according to the WRF model at the mouths of the Mzymta and Sochi rivers for the low-discharge year (2020) (left) and the high-discharge year (2021) (right). The scale of surface salinity fields is presented in psu. Wind speed is presented in m/s; Table S1: Periods (month, date), duration (day), maximum discharge peak (m$^3$/s), total discharge (km$^3$) of spring/spring-summer snowmelt freshets and flooding events in 2020–2021 for the Mzymta and Sochi rivers; Table S2: Periods of flooding events, wind roses at river mouths and distribution of Lagrangian particles (river water) during 2020–2021 years for the Mzymta and Sochi rivers. Refs. [32–38] are cited in Supplementary Materials.

**Author Contributions:** Conceptualization: A.O.; methodology: E.K., I.P., A.O., P.B. and V.F.; software: E.K., I.P, P.B. and V.F.; validation: E.K., I.P., P.B. and V.F.; formal analysis: E.K., I.P. and P.B.; investigation: E.K., I.P. and P.B.; resources: E.K., I.P., A.O., P.B. and V.F.; writing—original draft preparation: E.K., I.P. and A.O.; writing—review and editing: E.K., I.P., A.O., P.B. and V.F.; visualization: E.K. and I.P.; supervision: A.O. All authors have read and agreed to the published version of the manuscript.

**Funding:** This research was funded by Roshydromet, state assignment 121081600099-7 (numerical modeling of sea circulation); the Ministry of Science and Higher Education of the Russian Federation, themes FMWE-2021-0001 (study of river plumes) and FMWZ-2022-0003 (collecting and processing of hydrological data). Numerical calculations were performed using supercomputer resources of the Joint Supercomputer Center of the Russian Academy of Sciences.

**Data Availability Statement:** Data is contained within the article or Supplementary Materials.

**Conflicts of Interest:** The authors declare no conflict of interest.

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
