# Peer review of "Synoptic and Seasonal Variability of Small River Plumes in the Northeastern Part of the Black Sea"

_water, doi:10.3390/w15040721_

Round 1

Reviewer 1 Report (Previous Reviewer 2)

Dear authors,

I carefully passed through the resubmitted maniscript and hereby I report the peer review of the study presented within the manuscript. 

Specific issues have been improved compared to inituially submitted manuscript but still, several critical issues is present. I expect them to be resolved either elaborated along the review process.

1. The work is a combination od several methodolgical approaches. Initially, satelite images are taken, than discharge values are obtained by the web service, and finally several numerical models have been used to obtain velocity field and plume dispersion. If so, what is the guarantee that obtained and presented results are real, or do they correspond to real situation? This is the may concern since no validation of the results is presented. Hereby authors should improve the content with in situ observed results relevant for the validation of the results arose from this study. Hereby I explicitely focus to salinity and current velocity.

2. Grid convergence of the numerical model INMOM used is missed? Authors only referr to the number of elements but what is the evidence of the grid size appropriateness to the process modeled? Please conduct the grid convergence, can be done on a smaller area but strictly close to the river mouth.

3. Please refer to the validity and robustness of the findings? How valid they area, can the methodology be used to another similar site, are there oany obstacles to be faced to?

I suggest major review.

Author Response

Reviewer 2 Report (Previous Reviewer 1)

My comments on the previous version of the manuscript have been addressed properly. Several other comments are as follows: 

1. I feel that some results in the supplemental document should be added to the paper, for instance, the same contours as figure 6 but for year 2020. 

2. In the last paragraph of the introduction section and in the conclusions section, the authors described what they did and what they observed. However, the innovative contributions of the work, which are more than the satellite images and the simulation results and fill the knowledge gap, are not clearly presented.  Moreover, it is suggested to add some quantitive conclusions in the last section. 

3. Check the English writing: Abstract - "We use numerical modeling applied to the northeastern part of the Black Sea as the case study area with numerous small river plumes".  

Round 2

Reviewer 1 Report (Previous Reviewer 2)

No comments.

Reviewer 2 Report (Previous Reviewer 1)

My comments have been addressed properly. 

This manuscript is a resubmission of an earlier submission. The following is a list of the peer review reports and author responses from that submission.

Round 1

Reviewer 1 Report

Review report

Title: Synoptic and Seasonal Variability of Small River Plumes in the Northeastern Part of the Black Sea

Authors: Korshenko et al. 

Journal: Water

The authors employed numerical models to reconstruct the distribution of plumes in the northeastern part of the black sea. The examined quantities include the fields of surface salinity, plume areas, freshwater residence time, etc. The studied problem is challenging as it involves several different processes at different temporal and spatial scales. Comments are as follows:

1.     The finest spatial resolution in the INMOM simulation is 200 m, which cannot resolve the flow near the river mouth and is orders of larger than the flow scales related with the dispersion of Lagrangian particles. It is necessary to justify the capability of the models (the employed parameters) and the setup for such problems, and present the validation results by comparing with measurements if available. 

2.     Different models were employed. A schematic image or a flowchart showing the employed models, the procedure for solving the problem, and the way different models interact with each other, is needed for people to get an overview. 

3.     Two rivers were considered. Why not considering all rivers in the aera? 

4.     It is necessary to provide the governing equations employed in the INMOM and OpenDrift models. 

5.     It is suggested to change the title for section 2.5 to a descriptive one, instead of a general-purpose method. 

Reviewer 2 Report

Review of the manuscript

Synoptic and Seasonal Variability of Small River Plumes in the Northeastern Part of the Black Sea

by

Evgeniya Korshenko, Irina Panasenkova, Alexander Osadchiev, Pelagiya Belyakova and Vladimir Fomin

submitted for potential publication to Water MDPI

General comments:

Authors deal with the investigation of the plume refreshment effects of the Black Sea induced by the increased river inflow. The problem is absolutely relevant and up to date with a real site orientated demonstration.

From methodological point of view authors implemented different approaches starting form satellite observations, available river discharge values, Eulerian and Lagrangian based numerical models to obtain relevant results.

Besides the plume features arising from different hydrological conditions authors state high values of river discharge did not result in the largest area of the river plume. The freshwater residence time of plumes strongly depends on the duration, intensity and direction of the wind.

The manuscript structure suffers from scientific form especially in the way main conclusions are missed.

Below I attach several major and Minor comments authors may answer, refer either comment to:

Major comments:

Discussion section should demonstrate either the generality of the main findings or their limitation to the study area strictly. Without this the validity of the findings presented within the manuscript is unknown.

How INMOM works in a way of the output velocity field results? Is the velocity continuous or discontinuous since the finite grid used?

If the velocity from INMOM is used in Lagrangian analysis how the discontinuity ion velocity field has been overcomed?

I would appreciate to see more on the Lagrangian equation and especially advective and dispersive tensor components?

How the vertical stratification in the salinity is incorporated into the modeling part?

Minor Comments:

Table 1 – discharge is expressed in m3?

Reviewer 3 Report

Dear Authors,

I regret to inform you that your submission titled "Synoptic and Seasonal Variability of Small River Plumes in the Northeastern Part of the Black Sea"  needs a robust revision prior to be considered for publication.

As it is, I'm not able to express an opinion about the significance of its scientific content, because the structure of the paper, the construction of a lot of sentences and the quality of the English are very poor and what you mean remains very often obscure.

The abstract does not end with a sentence illustrating the novelty and general interest of your findings.

Methods leave in the reader a lot of doubts about the periods you really considered, why you used a 16.5 psu threshold for salinity and in which way WRF and INMOM models work.

Results are rather qualitative and, although in the title of sub-chapter 3.2 you stated that its topic is salinity, no salinity data are discussed therein; you simply declared that salinity is presented in Fig.5, but there is no discussion related to salinity in the following text. Moreover, as previously written, without describing WRF and INMOM models it is impossible to understand what kind of data are shown in the figure.

In paragraph 3.3 you finally attempted to describe the 16.5 psu salinity threshold (and note before, when requested), but what you mean remains obscure: you stated that this threshold corresponds to "....concentration of isohalines in the gradient zone....", but in a gradient zone there is a spatial variability of the related variable (salinity), so in which way this can be related to a fixed threshold value?

Finally, the discussion are a mere summary of what observed without any sentence highlighting the novelty, relevance and general interest of your findings.

Maybe your work contains interesting findings relevant for a broad audience, because applicable to other similar contexts worldwide, but the problem is that the overall quality of your presentation does not allow to discover this.

I attached a commented pdf with some other specific comments.
